# Ovine tricuspid annular dynamics and three-dimensional geometry during acute atrial fibrillation

**Paulina Kania-Olejnik[1], Marcin Malinowski [1,2] \*, Manuel K. Rausch[3], Tomasz A. Timek[2]**

**1** Department of Cardiac Surgery, School of Medicine in Katowice, Medical University of Silesia, Katowice, Poland, **2** Meijer Heart Center at Corewell Health, Grand Rapids, Michigan, United States of America, **3** Department of Aerospace Engineering & Engineering Mechanics, Department of Biomedical Engineering, Oden Institute for Computational Engineering and Science, The University of Texas, Austin, Texas, United States of America

\* marmal@interia.pl

## Abstract

**Data Availability Statement:** All data files are available from the OSF publicly available registry: https://osf.io/vkqm7/?view_only=655faa12eccc4327a20a826744c46bbf.

### Objectives

Long-standing atrial fibrillation (AF) may lead to tricuspid regurgitation (TR) and right ventricular dysfunction. However, the effect of acute AF on tricuspid annular (TA) dynamics and three-dimensional geometry is unknown.

### Methods

In eight adult sheep, sonomicrometry crystals were implanted around the tricuspid annulus and right ventricular free wall. Pressure transducers were placed in the right ventricle, left ventricle, and right atrium. After weaning from cardiopulmonary bypass and a period of hemodynamic stabilization, simultaneous sonomicrometry and hemodynamic data were collected in sinus rhythm (SR) and during experimental AF (400b/min right atrial pacing). Annular area, perimeter, dimensions, height, global and regional annular contraction, and strain were calculated based on cubic spline fits to crystal 3D locations.

### Results

Maximal TA area increased from $1084.9\pm273.9mm^2$ in SR to $1207.5\pm322.1mm^2$ during AF ($p = 0.002$). Anteroposterior diameter increased from $36.5\pm5.0mm$ to $38.4\pm5.5mm$ ($p = 0.05$). TA contraction decreased from $7\pm2\%$ in SR to $2\pm1\%$ in AF ($p = 0.001$). Anterior, posterior, and septal regional annular contraction decreased from $10\pm4\%$, $8\pm3\%$ and $6\pm2\%$ to $4\pm2\%$, $3\pm1\%$ and $2\pm1\%$ for SR and AF, respectively ($p<0.05$). AF perturbed systolic global annular strain (from $-6.52\pm1.74\%$ to $-2.78\pm1.79\%$; $p = 0.003$) and caused annular stretch. Annular height marginally decreased with AF from $5.8\pm1.9mm$ to $5.7\pm2.0mm$; $p = 0.039$.

**Funding:** The study was funded by internal funds from the Meijer Heart Center. The funders had no role in study design, data collection and analysis, decision to publish, or preparation of the manuscript.

**Competing interests:** The authors have declared that no competing interests exist.

## Conclusions

Acute experimental AF in healthy sheep was associated with TA dilation, flattening, and decreased total and regional annular contractility. These data may help elucidate the pathophysiology of functional TR associated with AF.

## Introduction

Functional tricuspid regurgitation (FTR) is a common valvular finding associated with increased mortality [1], yet historically, clinical attention has been focused on the associated left-sided valvular disease essentially ignoring the pathophysiology of FTR. This imbalance is rooted in a false belief that FTR carries no clinical consequence and that severe TR is uncommon. Functional TR has recently been identified as a "public health crisis" [2], being most frequently associated with left-sided valve dysfunction or pulmonary hypertension [3]. If untreated, functional TR is an independent risk factor for developing heart failure that eventually may lead to death [4, 5]. Recently, it was demonstrated that FTR arising from isolated annular dilation is not uncommon [6], spurring a novel dichotomous classification of atrial and ventricular FTR [7]. Atrial fibrillation (AF) is a common arrhythmia associated with enlargement of the atria and dilation of the atrioventricular annuli that may lead to significant mitral or TR [8, 9]. AF flattens and enlarges the mitral annulus, which has been demonstrated to be a strong predictor of mitral regurgitation [10]. Recent clinical data suggest that similar mechanisms may be at work on the right side of the heart [11], but tricuspid annular (TA) remodeling in AF is still incompletely understood. In the current ovine study, we set out to describe TA three-dimensional geometry, dynamics, and strains in acute experimental AF.

## Materials and methods

### Surgical preparation

All animals received humane care in compliance with the Principles of Laboratory Animal Care formulated by the National Society for Medical Research and the Guide for Care and Use of Laboratory Animals. The study protocol was approved by the Calvin University Institutional Animal Care and Use Committee (protocol No. 2016–07) in June 2016.

Eight healthy adult Dorset castrated male sheep (56±5kg) had an external right jugular intravenous catheter placed under local anesthesia with 1% lidocaine injected subcutaneously. Animals were then anesthetized with propofol (2–5 mg/kg administered intravenously), intubated, and mechanically ventilated. General anesthesia was maintained with inhalation of isoflurane (1%-2.5%). Fentanyl (5–20 mg/kg/min) was infused as additional maintenance anesthesia. A 4-Fr vascular access sheath was introduced for arterial blood pressure measurements through the left carotid artery. Animals were fully heparinized, and the right carotid artery and right internal jugular vein were exposed in preparation for cardiopulmonary bypass (CPB). The operative procedure was performed through a sternotomy, exposing the heart in a pericardial cradle. Caval snares were placed, and the superior and inferior vena cava were cannulated with a multistage venous cannula (BioMedicus™ 21Fr, Medtronic, MN) via the right jugular vein. While on CPB and with the heart beating, both cava were snared, the right atrium opened, and six 2-mm sonomicrometry crystals (Sonometrics Corp, London, Ontario, Canada) were implanted with 5–0 polypropylene suture around the tricuspid annulus. One crystal was implanted at each commissure, and an additional crystal was equidistant between the commissures defining three annular regions (anterior, posterior, and septal; Fig 1). Four additional crystals were implanted on the right ventricular epicardium along the right ventricular free wall, with a fifth crystal at the right ventricular apex. An

electrocardiogram (ECG) electrode connected to the sonomicrometry system was sutured to the right ventricular free wall. Pressure transducers (PA4.5-X6; Konigsberg Instruments Inc, Pasadena, CA) were placed in the left ventricle and right ventricle through the apex, with an additional pressure transducer placed in the right atrium.

After completion of crystal implantation, the atriotomy was closed, and the animal was weaned from CPB. Animals were allowed to stabilize for 30 minutes to achieve steady-state hemodynamic parameters after weaning from CPB. Simultaneous sonomicrometry, hemodynamic, and epicardial echocardiographic data were first collected in SR and then during experimental AF (burst atrial pacing at 400/min). All animals were studied under open-chest experimental conditions. After the experiment, the animals were euthanized by intravenously administering sodium pentothal (100mg/kg) and potassium chloride intravenous bolus (80 mEq). The heart was excised, and the proper placement of crystals was confirmed.

## Data acquisition

As previously described, all sonomicrometry data were acquired using a Sonometrics Digital Ultrasonic Measurement System DS3 (Sonometrics Corp, London, Ontario, Canada) [12]. Data were obtained at 128 Hz with simultaneous left ventricular pressure, right ventricular pressure, central venous pressure, and ECG recordings. Data from 10 consecutive cardiac cycles were averaged during SR and during AF. All sonomicrometry recordings were analyzed with custom MATLAB code (MathWorks, Natick, MA). All values were calculated at end-systole (ES), end-diastole (ED), and during maximal and minimal TV area time. ED was defined as the time of the beginning of positive deflection in ECG voltage (R wave), whereas ES was determined as the time of maximum negative dp/dt of left ventricular pressure.

## Data analysis

**Tricuspid annular geometry and dynamics.** TA area and perimeter were calculated based on the spline fit of annular crystals as previously described [13, 14]. Briefly, piecewise cubic Hermitian splines that minimalize the distance to the crystals and meet a minimum mathematical smoothness requirement were obtained. The spline is a function of the arc-length parameter and time. Septolateral (S-L) annular dimension was calculated as the distance between crystals #2 and #6; anteroposterior (A-P) annular dimension as the distance between crystals #1 and #4. Intercommissural distances were calculated as the respective individual distances between crystals #1, #3, and #5 (Fig 1). Annular 3D geometry was represented by annular height, defined as the plane-normal distance between the two maximally displaced annular crystals above and below the best-fit annular plane. Annular height to commissural width ratio (AHCWR) was calculated as the ratio of maximal (diastolic) and minimal (systolic) displacement of annular crystals from the annular plane (height) to its appropriate A-P diameter. Annular area contraction was characterized as the percentage difference between maximal and minimal annular area during the cardiac cycle ($[Amax - A_{min}]/A_{min}$ x 100%). The regional annular contraction was defined as the percentage difference between maximal and minimal regional perimeter. The presystolic annular area contraction was defined as the percentage of total area reduction occurring between late diastolic maximum and end-diastole. Right ventricle volume was calculated using the convex hull method based on crystal coordinates.

**Annular strains.** TA strains were calculated as previously described [13, 15]. The strain is calculated as a relative measure of displacement; thus, annular deformation was calculated relative to a reference configuration. For the within cardiac cycle global and regional strain analysis, maximal TV area time (MAX) was chosen as the reference configuration, assuming it is a state of minimal annular stress. Interventional strain analysis included averaged SR MAX,

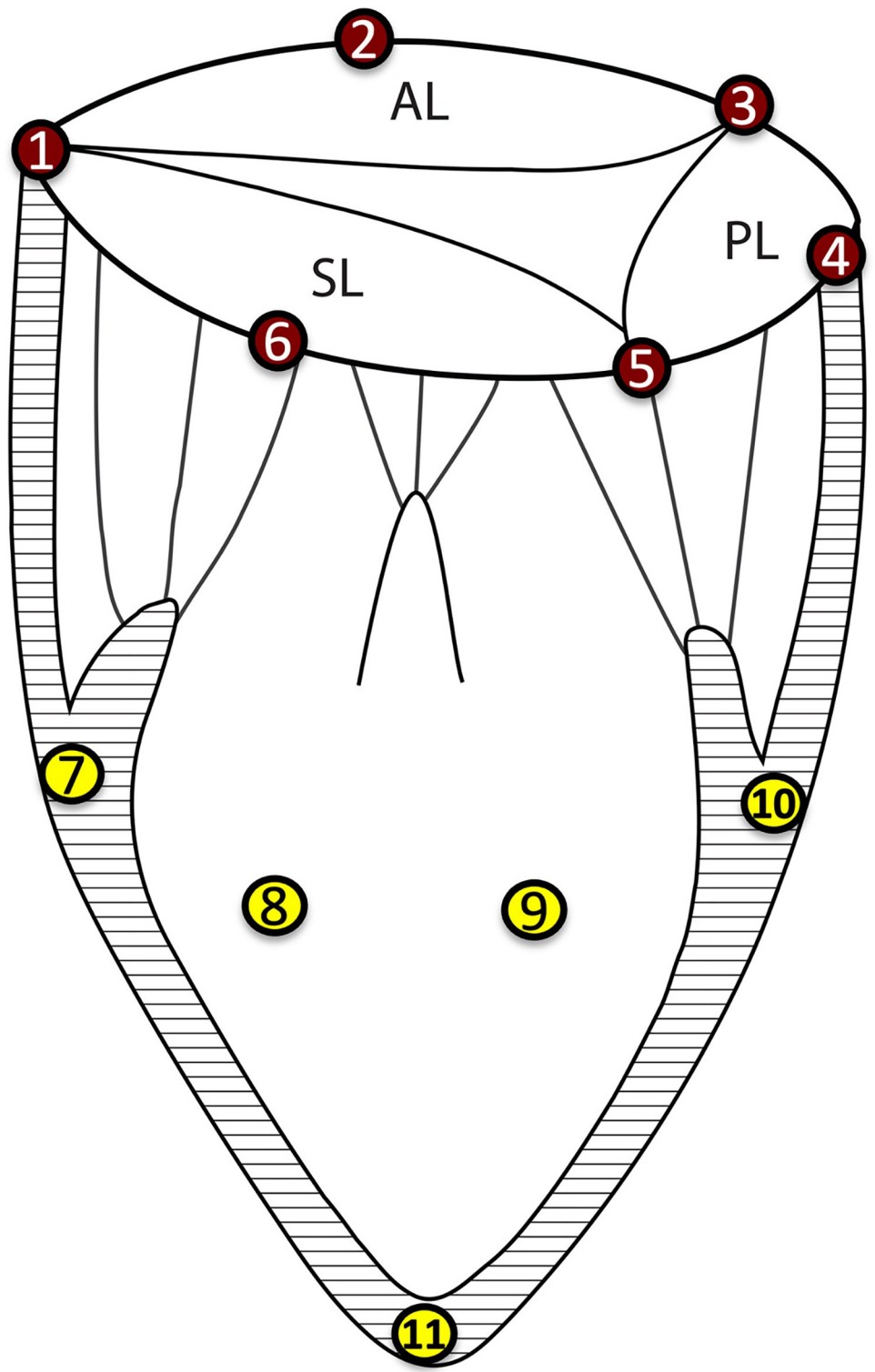

**Fig 1. The location of the sonomicrometry crystals implanted on the tricuspid annulus, the right ventricle free wall, and the right ventricle apex.** *AL*, anterior leaflet; *PL*, posterior leaflet; *SL*, septal leaflet.

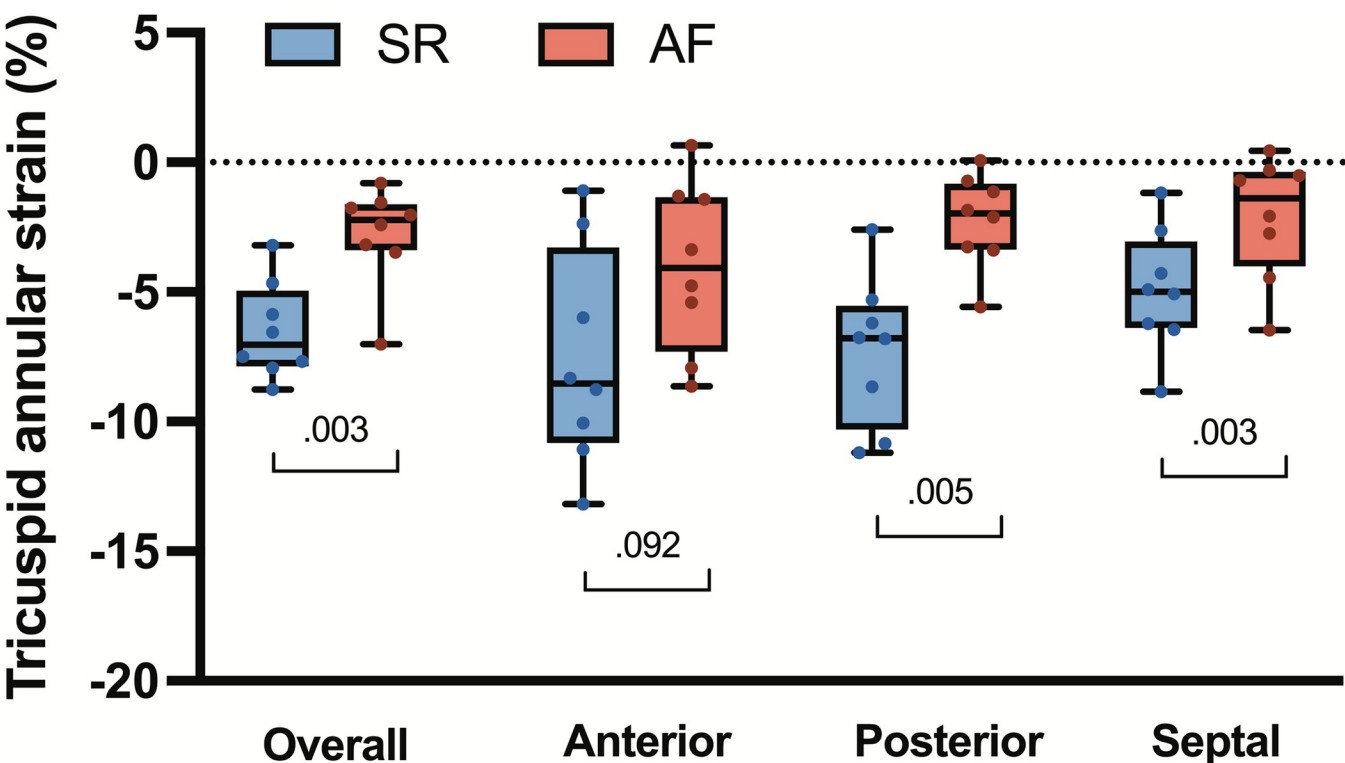

**Fig 2. Global and regional average systolic tricuspid annular cardiac strains.** Values are calculated with reference to the configuration at maximal valve area time. The *upper* and *lower borders* of the box represent the upper and lower quartile. The *middle horizontal line* represents the median. The *upper* and *lower whiskers* represent the maximum and minimum values. *P* values from paired t-test. *SR*, sinus rhythm; *AF*, atrial fibrillation.

end-diastole, end of isovolumic contraction, end-systole, end of isovolumic relaxation, and minimal valve area times as the reference states for respective time points during AF. Therefore, cardiac strain reflects the deformation of the annulus throughout the cardiac cycle, and interventional strain reflects the deformation of the annulus induced by rhythm change. Precisely, the Green-Lagrange strain was calculated along the entire annulus for each animal and later displayed on a spline representation of the population-averaged annulus for each group. Global and regional averaged TA strains were calculated for the entire annulus and anterior, posterior, and septal regions by averaging them along the respective regions (Fig 2).

## Statistical analysis

Data are presented as mean±one standard deviation. Normality assumptions were tested with the Shapiro-Wilk test. The within-heart cycle comparisons were performed using Student's t-test for dependent observations or the Mann-Whitney test when the normality test failed. The variables compared between SR and AF conditions were similarly tested. Values of p <0.05 were considered significant. GraphPad Prism 9.3 (GraphPad Software, San Diego, CA) was utilized for statistical analysis.

## Results

### Hemodynamic characteristics

Hemodynamic parameters for SR and atrial fibrillation are presented in Table 1. To summarize, right ventricle volume and right and left ventricular systolic pressures decreased in atrial fibrillation. There was no change in right atrial pressure.

**Table 1. Hemodynamic parameters.**

| (n = 8) | Sinus rhythm | Atrial fibrillation | P value |
|---|---|---|---|
| **HR (min$^{-1}$)** | 112±20 | 196±43 | <0.001 |
| **RVP max. (mmHg)** | 30±6 | 23±7 | <0.001 |
| **RV EDP (mmHg)** | 12±10 | 8±5 | 0.175 |
| **RVP max dp/dt (mmHg/s)** | 594±177 | 388±169 | 0.001 |
| **RV EDV (ml)** | 70±18 | 63±16 | 0.001 |
| **RAP (mmHg)** | 15±6 | 13±5 | 0.181 |
| **LVP max (mmHg)** | 102±11 | 58±13 | <0.001 |
| **LV EDP (mmHg)** | 14±6 | 11±6 | 0.426 |
| **LVP max dp/dt (mmHg/s)** | 2136±629 | 1166±472 | <0.001 |

HR = heart rate, RV = right ventricle, RAP = right atrial pressure, LV = left ventricle, EDP = end diastolic pressure, EDV = end diastolic volume

## Tricuspid annular geometry and dynamics

The maximal TA area in diastole was 1084.9±273.9mm$^2$ in SR and increased to 1207.5 ±322.1mm$^2$ in atrial fibrillation ($p$ = 0.002). The minimal annular area during systole was 1056.2±275.8mm$^2$ in the SR and 1183.8±315.8mm$^2$ in atrial fibrillation ($p$ = 0.002). TA area throughout the cardiac cycle in SR and AF is shown in Fig 3. Maximal TA perimeter increased from 115.7±11.9mm in SR to 118.2±12.3mm in AF; $p$ = 0.002. Similar changes were observed with minimal TA circumference; 107.3±12.1mm and 114.7±12.9mm for SR and AF,

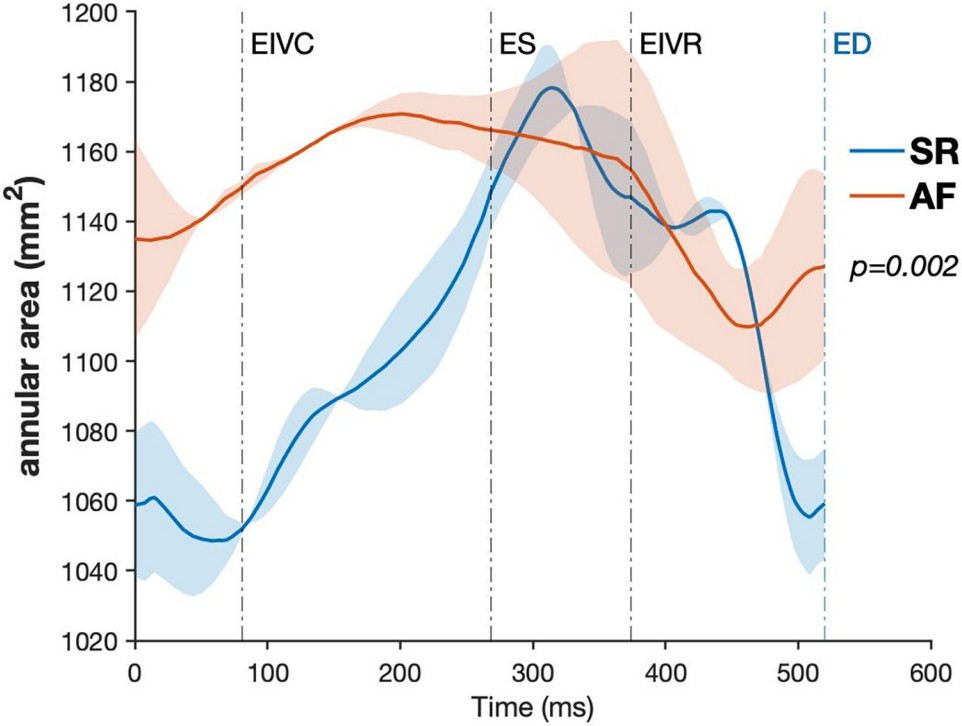

**Fig 3. The annular area throughout the cardiac cycle in SR and AF.** Data are presented as mean (solid line) ± standard error (shaded area). *SR*, sinus rhythm; *AF*, atrial fibrillation; *ED*, end-diastole; *ES*, end-systole; *EIVC*, end of isovolumic contraction; *EIVR*, end of isovolumic relaxation.

**Table 2. Regional tricuspid annular perimeter.**

| (n = 8) | | Sinus rhythm | Atrial fibrillation | P value |
|---|---|---|---|---|
| **Anterior annulus (mm)** | | | | |
| Overall | maximal | 33.3±3.5 | 35.0±3.8 | 0.063 |
| | minimal | 29.9±3.4 | 33.3±4.1 | 0.006 |
| #1–2 | maximal | 16.6±2.0 | 16.8±1.8 | 0.760 |
| | minimal | 14.4±1.8 | 15.5±1.6 | 0.116 |
| #2–3 | maximal | 17.1±3.1 | 18.6±3.9 | 0.264 |
| | minimal | 15.1±3.3 | 17.7±3.9 | 0.052 |
| **Posterior annulus (mm)** | | | | |
| Overall | maximal | 35.8±6.8 | 35.7±6.3 | 0.745 |
| | minimal | 32.8±6.5 | 34.6±6.3 | 0.013 |
| #3–4 | maximal | 18.7±4.5 | 18.3±3.8 | 0.594 |
| | minimal | 16.6±4.1 | 17.3±3.7 | 0.233 |
| #4–5 | maximal | 17.7±4.5 | 17.7±4.6 | 0.749 |
| | minimal | 16.0±4.2 | 17.1±4.4 | 0.010 |
| **Septal annulus (mm)** | | | | |
| Overall | maximal | 47.3±6.3 | 48.4±6.7 | 0.006 |
| | minimal | 43.8±5.5 | 46.6±6.5 | <0.001 |
| #5–6 | maximal | 21.4±4.2 | 22.2±4.5 | 0.054 |
| | minimal | 18.5±3.9 | 20.9±4.7 | <0.001 |
| #6–1 | maximal | 26.7±4.5 | 26.9±5.0 | 0.617 |
| | minimal | 24.9±4.1 | 25.5±4.6 | 0.259 |

respectively ($p$<0.001). The regional changes in annular circumference are shown in Table 2. The S-L, A-P, and inter-commissural annular distances are presented in Table 3. Annular A-P diameter increased during atrial fibrillation along with minimal inter-commissural, anteroseptal, anteroposterior, and posteroseptal distances. Maximal and minimal S-L diameters remained unaffected by AF.

TA area change (contraction) during the cardiac cycle was 7±2% in SR and dropped to 2 ±1% in atrial fibrillation ($p$ = 0.001). The presystolic annular area contraction was significantly ($p$ = 0.021) lower in AF (55±30%) than in SR (80±14%). Induced atrial fibrillation affected regional contraction with a significant reduction in each annular region (Fig 4). The contraction in the anterior annular region (#1-#3) changed from 10±4% in SR to 4±2% in atrial fibrillation ($p$ = 0.018). Similarly, contraction for septal annular region (#3-#5) was 6±2% vs. 2±1% ($p$<0.001), and in the posterior annulus (#5-#1) 8±3% vs. 3±1% ($p$ = 0.003) for SR and AF, respectively. A typical nonplanar saddle-shaped annulus morphology observed in SR was affected by experimental AF. This is demonstrated in a significant change of annular height and annular height-to-commissural width ratio (AHCWR). The maximal annular height decreased with AF (from 5.8±1.9 to 5.7±2.0mm; $p$ = 0.039), and AHCWR diminished from 18 ±7% in SR to 17±7% in AF ($p$<0.001).

## Annular strains

The global average systolic cardiac annular strain was -6.52±1.74% in SR, and its magnitude decreased in atrial fibrillation to -2.78±1.79%; ($p$ = 0.003). The changes in regional cardiac strain within the cardiac cycle for both SR and after AF are shown in Fig 5. AF significantly perturbed the course of the annular strain in all regions. However, AF did not considerably affect the systolic anterior annular cardiac strain but reduced systolic strain in the septal and

**Table 3. Tricuspid annular dimensions.**

| (n = 8) | Sinus rhythm | Atrial fibrillation | P value |
|---|---|---|---|
| **S-L diameter (mm)** | | | |
| maximal | 32.2±4.4 | 32.7±4.0 | 0.177 |
| minimal | 29.3±5.2 | 31.0±3.9 | 0.06 |
| **A-P diameter (mm)** | | | |
| maximal | 36.5±5.0 | 38.4±5.5 | 0.05 |
| minimal | 32.3±5.4 | 35.5±6.1 | <0.001 |
| **C-C 1–3 (mm)** | | | |
| maximal | 33.3±3.5 | 35.0±3.8 | 0.063 |
| minimal | 29.9±3.4 | 33.3±4.1 | 0.006 |
| **C-C 3–5 (mm)** | | | |
| maximal | 35.8±6.8 | 35.7±6.3 | 0.745 |
| minimal | 32.8±6.5 | 34.6±6.3 | 0.013 |
| **C-C 5–1 (mm)** | | | |
| maximal | 47.3±6.3 | 48.4±6.7 | 0.006 |
| minimal | 43.8±5.5 | 46.6±6.5 | <0.001 |

Values are presented as mean±standard deviation. Crystal #1 = anteroseptal commissure; crystal #3 = anteroposterior commissure; crystal #5 = posteroseptal commissure. *S-L*, septo-lateral; *A-P*, antero-posterior; *C-C*, intercommisural distances.

posterior regions (Fig 6). Fig 6 presents the color map of average annular interventional strains. With SR as a reference state (green color), AF increased global interventional strain at ES (5.3±4%; $p = 0.01$) and ED (7.1±3%; $p<0.001$), indicating overall annular stretch. In AF, the largest relative stretch was observed in the anterior region (10.5±11.4% at ES [$p = 0.044$] and 11.1±9.4% at ED [$p = 0.017$]). For the posterior region, these values were 2.9±2.19% and 4.9±3.5 at ES and ED, respectively ($p = 0.008$). Interestingly, annular stretch was also observed in the septal region (ES 3.8±4.8% [$p = 0.076$] and ED 6.2±3.0% [$p = 0.001$]). This stretch was mostly visible in the septal region adjacent to the posterior commissure (ES 6.2±5.2% [$p = 0.016$] and ED 11.0±6.4% [$p = 0.003$]) whereas the septal part close to the anterior commissure was relatively static (ES 1.8±4.7% [$p = 0.335$] and ED 2.3±4.0 [$p = 0.168$]). In summary, induction of experimental atrial fibrillation caused significant annular stretch relative to the SR.

## Discussion

The results of our study reveal meaningful TA changes with isolated acute experimental AF. Similar to other investigators [16], our data demonstrate that in the absence of concomitant pulmonary hypertension or co-existing left-sided heart disease, lone AF may lead to TV annular dilation. Our experiment revealed AF to be associated with significant increase in annular area and decrease in annular contractility as well as annular stretch, especially in the septal region. We also observed changes in 3D annular geometry as the annulus became flatter, and its saddle shape became more planar. Our findings shed light on the poorly understood dynamic behavior of the TA in AF and corroborate clinical findings in patients with atriogenic functional TR [17, 18]. To our best knowledge, this is the first work to date investigating the isolated effect of lone AF on the tricuspid annulus.

The clinical influence of atrial fibrillation on atrio-ventricular valve competence was initially studied by Gertz et al. [10] who described AF-related mitral regurgitation in patients

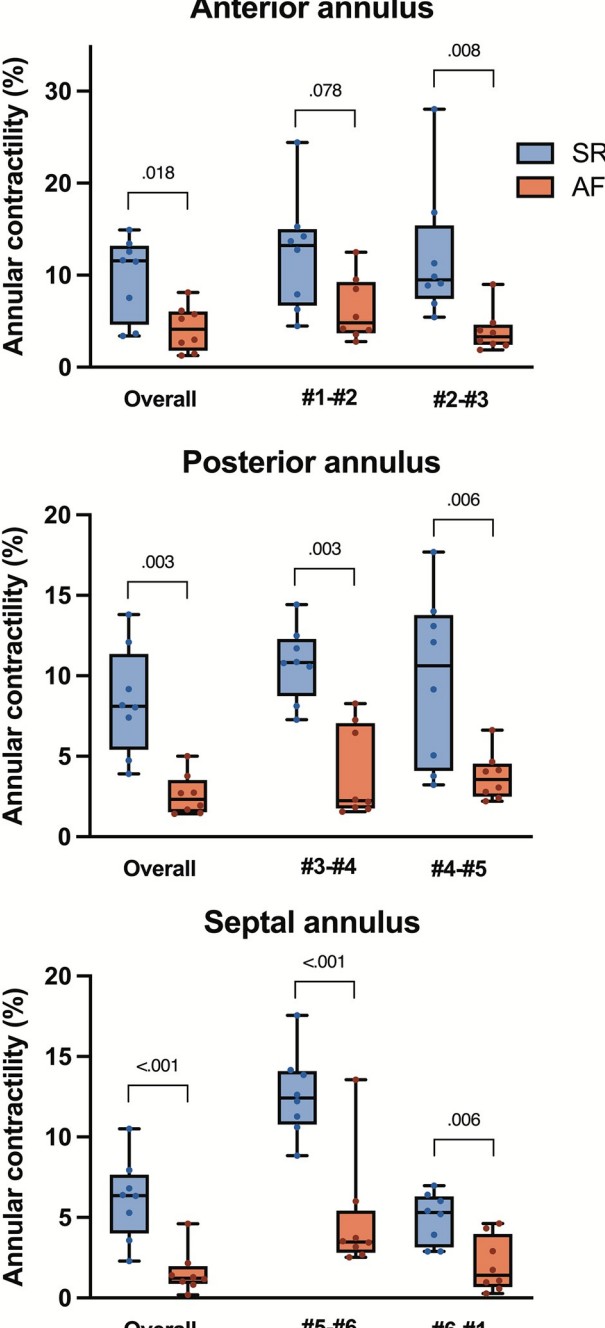

**Fig 4. Regional tricuspid annular contractility in the SR and during induced atrial fibrillation.** Contractility was calculated as the percent difference between maximal and minimal regional perimeter. Crystal numbers (#1–6) as depicted in Fig 1. The *upper* and *lower borders* of the box represent the upper and lower quartile. The *middle horizontal line* represents the median. The *upper* and *lower whiskers* represent the maximum and minimum values *Top;* anterior annulus (crystal #1, #2, #3). *Middle*, posterior annulus (crystals #3, #4, #5). *Bottom*, septal annulus (crystals #5, #6, #1). *SR*, sinus rhythm; *AF*, atrial fibrillation.

with associated mitral annular dilation and structurally normal leaflets. One-year after AF ablation, mitral annulus (MA) dimensions showed significant reverse remodeling with institution of SR establishing the physiologic link between rhythm and annular remodeling. More

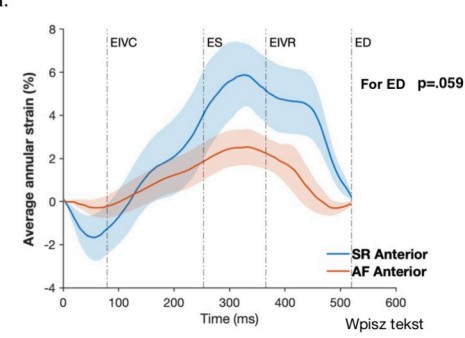

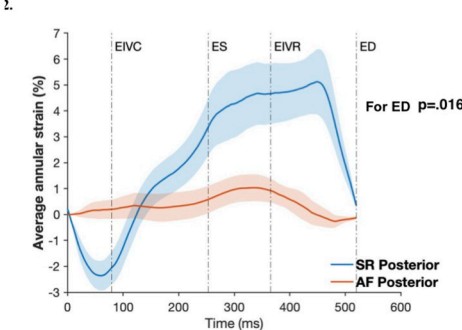

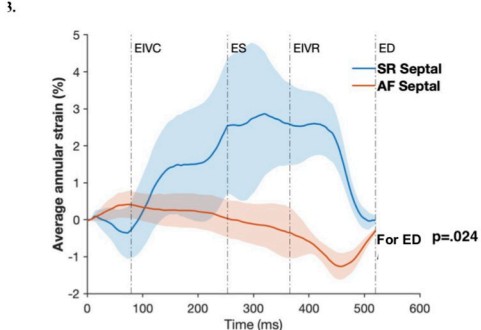

**Fig 5. Regional cardiac strain changes during cardiac cycle.** Data are presented as mean (solid line) ± standard error (shaded area). *SR*, sinus rhythm; *AF*, atrial fibrillation; *ED*, end-diastole; *ES, end-systole; EIVC,* end of isovolumic contraction; *EIVR*, end of isovolumic relaxation. 1. Anterior annular strain; 2. Posterior annular strain; 3. Septal annular strain.

detailed MA changes were described by Kim and colleagues, who demonstrated that atrial fibrillation caused not only annular dilation but also annular flattening [19]. Our prior ovine data illustrated the need for coordinated atrioventricular contraction to achieve timely and effective mitral valve closure [20] and support these clinical findings and the reduced presystolic dynamic behavior of the annulus during AF observed by others [21].

The tricuspid and the mitral annulus have similar geometry resembling a saddle-like shape with high points in near the anteroseptal commissure and mid-posterior annulus. The nonplanar shape of TA has been confirmed in animals and humans utilizing sonomicrometry, 3D echocardiography, and magnetic resonance imaging [15, 22–24]. Fukuda's study demonstrated that TA became larger and flatter with an increased A-P diameter in patients with TR

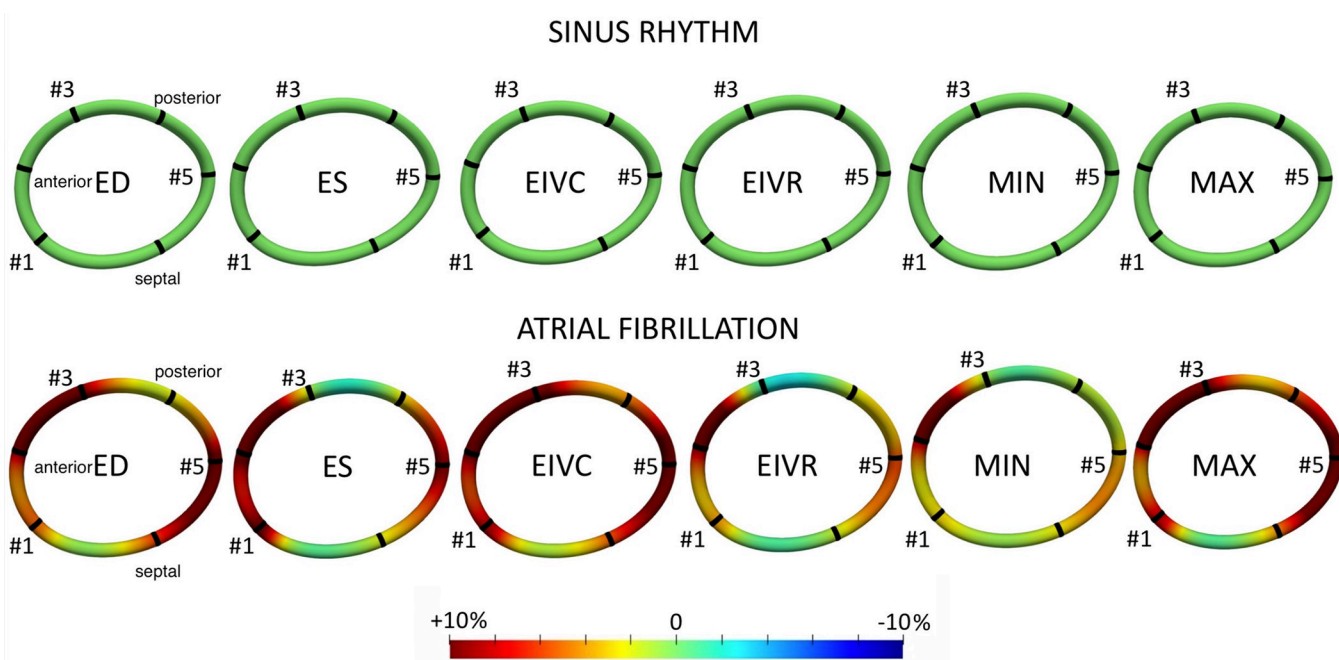

**Fig 6.** Average tricuspid annular strains in atrial fibrillation (lower panel) with reference to sinus rhythm configuration (upper panel). The metric is shown on the average annular spline. The positive strain values indicate relative annular stretch, and the negative values annular compression. Crystal 1, 3, and 5 represent anteroseptal, anteroposterior, and posteroseptal commissures, respectively. *ED*, end-diastole; *ES*, end-systole; *EIVC*, end of isovolumic contraction; *EIVR*, end of isovolumic relaxation; *MAX*, maximal valve area time; *MIN*, minimal valve area time.

[25]. Similar annular findings were observed in our study corroborating those clinical data and establishing a potential pathophysiologic link between AF and TR. However, our experiment's acute design with short duration of AF and limited AF-related annular dilation was insufficient to induce detectable TR in our ovine model. In contrast to Fukuda's data, we observed reduced TA contractility with AF and postulate that in long-standing AF annular contractility may be preserved as a compensatory effect. Furthermore, in patients with FTR, the TA was found to be dilated in the septolateral (SL) direction resulting in a more circular shape which we did not observe in our experimental AF preparation. Whether this is due to species, duration, or mechanism of TR differences remains unclear. Physiologically, coordinated atrial contraction reduces the annular size before ED to prepare the valve for closure [26], with subsequent ventricular systole completing valve closure. The degree of this pre-systolic annular contraction was found to be lower in AF than in SR in our study, and this physiologic mechanism, when disturbed in AF, could be associated with initiation and progression tricuspid insufficiency in patients with long-standing AF.

More recently, Naser and colleagues specifically compared TA function and dynamics in AF and SR patients [27]. Similar to our results, the authors demonstrated larger maximal TA area and circumference in AF patients and reduced TA area change during the cardiac cycle. These data corroborate the blunted TA dynamics observed in our study with experimental AF. The authors reported lower RA reservoir and RV free-wall longitudinal strains, but annular strains were not measured. We focused solely on annular strain, and although its relationship to atrial and ventricular strains is at this time unclear, clinical studies suggest that reduction in annular strain may be linked to strain decrease in the atrium and ventricle [28, 29]. Unfortunately, mitral and TA strains have not been studied extensively, and only a handful of reports analyzed MA strains in sheep in SR [30] and during ischemic mitral regurgitation [31]. Others

have focused on left ventricular longitudinal and circumferential or atrial strains using two-dimensional speckle tracking echocardiography [29]. To our best knowledge, TA strains in AF have not been described previously. The native ovine annulus cardiac cycle strains in SR found in the current study agree with our previous report [13]. We additionally discovered that TA strain varies considerably across the annulus in SR, and this process is further accentuated in AF.

In conclusion, in healthy adult sheep, experimental AF was associated with increased TA area and reduced annular dynamics, which may contribute to atrial FTR seen clinically. It should be emphasized that in this acute experimental study, a hypothetical link between alterations in annular geometry induced by AF and TR is suggested, yet this requires further study and confirmation. However, our results shed light on the potential pathogenesis of AF-related TR with clinical implications for surveillance and treatment. Chronic AF studies in animal models are needed to define the relationship between annular size and dynamics and FTR.

## Limitations

The current study's results must be interpreted in light of several limitations. This was an acute animal study, and clinical extrapolation of the results must be done with extreme caution. Our experiment was performed in open-chest healthy sheep under anesthesia, but we have previously shown that the anesthesia may affect TA dynamics but does not alter annular geometry [32]. The experiment was conducted on animals with normal hearts without atrial remodeling that is seen with long-standing AF. The induced AF was experimental and may differ in its effect from native AF seen clinically. In particular, we did not observe RA pressure change nor had a chance to access the TA under various heart rate responses. However, using healthy animals permitted the assessment of the direct effect of induced atrial fibrillation on annular dynamics and strain without potentially confounding factors. Moreover, we did not measure right atrium diameters, which are closely related to annular geometry and hemodynamics in AF.

## Supporting information

**S1 Video. Tricuspid annular strain changes throughout the cardiac cycle during sinus rhythm.**
(MP4)

**S2 Video. Tricuspid annular strain changes throughout the cardiac cycle during acute experimental atrial fibrillation.**
(MP4)

## Author Contributions

**Conceptualization:** Tomasz A. Timek.

**Data curation:** Marcin Malinowski.

**Formal analysis:** Paulina Kania-Olejnik, Marcin Malinowski.

**Funding acquisition:** Tomasz A. Timek.

**Investigation:** Marcin Malinowski, Tomasz A. Timek.

**Methodology:** Manuel K. Rausch, Tomasz A. Timek.

**Resources:** Marcin Malinowski.

**Software:** Manuel K. Rausch.

**Supervision:** Tomasz A. Timek.

**Validation:** Manuel K. Rausch.

**Writing – original draft:** Paulina Kania-Olejnik.

**Writing – review & editing:** Marcin Malinowski, Manuel K. Rausch, Tomasz A. Timek.

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
