## [Decision Letter · Decision Letter 0]

5 Jul 2024

PONE-D-24-14958Ovine Tricuspid Annular Dynamics and Three-Dimensional Geometry During Acute Atrial FibrillationPLOS ONE

Dear Dr. Malinowski,

Thank you for submitting your manuscript to PLOS ONE. After careful consideration, we feel that it has merit but does not fully meet PLOS ONE’s publication criteria as it currently stands. Therefore, we invite you to submit a revised version of the manuscript that addresses the points raised during the review process.

We look forward to receiving your revised manuscript.

Kind regards,

Ibrahim Marai, MD

Academic Editor

PLOS ONE

 [The study was funded by an internal funds from the Meijer Heart Center. ].  

Reviewers' comments:

Reviewer's Responses to Questions

**Comments to the Author**

1. Is the manuscript technically sound, and do the data support the conclusions?

Reviewer #1: Yes

Reviewer #2: Yes

2. Has the statistical analysis been performed appropriately and rigorously? 

Reviewer #1: N/A

Reviewer #2: Yes

3. Have the authors made all data underlying the findings in their manuscript fully available?

Reviewer #1: Yes

Reviewer #2: Yes

4. Is the manuscript presented in an intelligible fashion and written in standard English?

Reviewer #1: Yes

Reviewer #2: Yes

5. Review Comments to the Author

Reviewer #1: Main limitation is the small sample size

Please discuss the results in the context of transcatheter tricuspid valve replacement/intervention

Main limitation is the small sample size

Please discuss the results in the context of transcatheter tricuspid valve replacement/intervention

Main limitation is the small sample size

Please discuss the results in the context of transcatheter tricuspid valve replacement/intervention

Main limitation is the small sample size

Please discuss the results in the context of transcatheter tricuspid valve replacement/intervention

Reviewer #2: In this manuscript, titled “Ovine Tricuspid Annular Dynamics and Three-Dimensional Geometry During Acute Atrial Fibrillation” by Paulina Kania-Olejnik et al., the authors investigate the dynamic changes in tricuspid annular 3-dimensional (3D) geometry using an ovine model and provide the hemodynamic implications of acute atrial fibrillation (AF) on the tricuspid valvular dynamics and function. Below are my comments for the authors.

[1] From a clinical perspective, one of the most important issues is the experimental AF, which adopted rapid atrial pacing (400 bpm). The main hemodynamic feature of AF would be the lack of meaningful atrial contraction, which can result in different responses of the ventricle (i.e., rapid vs. controlled [or moderate] vs. slow ventricular rates). In the present study, the HR increased from 112±20 bpm in SR to 196±43 bpm in AF, and the RV EDV decreased from 70±18 ml in SR to 63±16 ml in AF, while there was no significant difference in RA pressure. Thus, it seems that the experimental acute AF in the present study resulted in a rapid ventricular response, reflecting only a limited aspect of the various hemodynamic consequences of AF on atrial/ventricular/annular dynamics. Although I acknowledge that the authors provided some explanation regarding this issue in the Limitations section, I think it needs further elaboration in the Discussion section.

[2] Discussion section (page 15, lines 292 – 294): The sentence, “Physiologically, coordinated atrial contraction reduces the annulus before ED to prepare the valve for closure with subsequent ventricular systole completing valve closure”, seems a bit confusing. Given the contents of the referenced article in this sentence (Tsakiris AG et al. Circ Res. 1975;36:43-8), I wonder whether the authors aimed to describe the reduced annular size (or area) during coordinated atrial contraction.

[3] The lack of measurement of RA size (diameter or volume), the lack of significant difference in RA pressure between SR and AF, and the reduced RV EDV under rapid atrial pacing (experimental AF) need to be further emphasized in the Discussion or Limitations section. Considering that the dilation of the RA and RV, as well as the increase in RA pressure, are known to be the main drivers of the development and progression of tricuspid regurgitation, these points are crucial. It could also be assumed that the lack of detectable TR in the authors’ ovine experimental model could be due to the experimental design, which induced a rapid ventricular response but did not cause dilation of the RA and RV.

[4] Figure 6: I recommend the authors add indications for the orientation of the tricuspid annulus (i.e., Anterior, Septal, and Posterior) in the figure, as done in their previous publication (Figure 4 in the study by Malinowski et al. J Thorac Cardiovasc Surg 2019;157:1452-61). The current indications (#1, #3, and #5) are sufficient to find the orientation but could be confusing for some readers if additional descriptions (i.e., Anterior, Septal, and Posterior) are not provided.

[5] Were there any specific reasons for describing the tricuspid annular area change (contraction) without providing a decimal place? It seems a bit awkward as most other measurements are provided to the first decimal place, but the “tricuspid annular area change” values are not.

[6] Minor comments

• Abstract: Please review the abbreviation for AF in the Abstract (Results section: Atrial fibrillation perturbed systolic global annular strain…).

• Abstract: Please check the number of standard derivation for the post-procedural systolic global annular strain (-2.78±-2.88%).

• Methods: In the Methods section of the main text, please use a consistent form of describing the states of the USA for devices/software used in the present study (i.e., Medtronic, Minneapolis, “MN”; Konigsberg Instruments Inc, Pasadena, “Calif”; MathWorks, Natick, “Mass”; GraphPad Software, San Diego, “CA”).

• In the main text, “Figure 3 (page 7)” appears earlier than “Figure 2 (page 9)”.

• I recommend the authors use a consistent form for describing the p-value (in some figures, the p-values are described as 0.XXX, but in other figures as .XXX).

• Please review the abbreviations for SR and AF throughout the manuscript (they appear in their abbreviated forms in the Results section [page 12 – Annular Strains], but in full names in other parts of the manuscript). Additionally, please check the abbreviation for the tricuspid annulus and functional tricuspid regurgitation, as they sometimes appear in their abbreviated forms (TA and FTR), but sometimes in their full name (tricuspid annulus, functional tricuspid regurgitation, or functional TR).

• Discussion (page 14): When citing the study by DH Kim et al. (Ref #19: JACC Cardiovasc Imaging. 2019;12:665-677), please use the family name of the author (“Kim”), rather than the given name (“Dae-Hee”).

6. PLOS authors have the option to publish the peer review history of their article (what does this mean?). If published, this will include your full peer review and any attached files.

Reviewer #1: No

Reviewer #2: No

---

## [Author Response · Author response to Decision Letter 0]

10 Sep 2024

#1 Reviewer’s comments: 

[1] Main limitation is the small sample size.

Author's Responses to Questions: 

Indeed, objectively, the study involves a small sample size, and this possess is a risk of a type II error. However, taking into account that this is an experimental study on large animals with a group of subjects and controls, we believe that this work is reliable and contributes new knowledge. Indeed that it is one in a series of studies with a similar number of animals

- Jazwiec T et al.. Tricuspid valvular dynamics and 3-dimensional geometry in awake and anesthetized sheep. J Thorac Cardiovasc Surg. 2018 Oct;156(4):1503-1511

- Malinowski M,et al. Tricuspid leaflet kinematics after annular size reduction in ovine functional tricuspid regurgitation. J Thorac Cardiovasc Surg. 2022 Dec;164(6):e353-e366;

- Malinowski M et al.Tricuspid Annular Geometry and Strain After Suture Annuloplasty in Acute Ovine Right Heart Failure. Ann Thorac Surg. 2018 Dec;106(6):1804-1811. 

Additionally, this number of animals was previously used by others when studying mitral valve: 

- Nguyen TC et al. The effect of pure mitral regurgitation on mitral annular geometry and three-dimensional saddle shape. J Thorac Cardiovasc Surg. 2008 Sep;136(3):557-65. doi: 10.1016; 

- Gorman JH 3rd et al.. The effect of regional ischemia on mitral valve annular saddle shape. Ann Thorac Surg. 2004 Feb;77(2):544-8. doi: 10.1016.

[2]: Please discuss the results in the context of transcatheter tricuspid valve replacement/intervention.

Author's Responses to Questions: 

Endovascular techniques for tricuspid valve repair or replacement are promising new technologies. Based on our results, we believe that a transcatheter tricuspid valve replacement approach, such as implantation of the commercially available Edwards Evoque valve, has strong potential to eliminate tricuspid regurgitation secondary to atrial fibrillation. Due to the fact that we did not examine the tricuspid valve leaflets, we cannot comment on the edge to edge method.

#2 Reviewer’s comments

We would like to thank Reviewer 2 for his comments and suggestions.

.

[1]: From a clinical perspective, one of the most important issues is the experimental AF, which adopted rapid atrial pacing (400 bpm). The main hemodynamic feature of AF would be the lack of meaningful atrial contraction, which can result in different responses of the ventricle (i.e., rapid vs. controlled [or moderate] vs. slow ventricular rates). In the present study, the HR increased from 112±20 bpm in SR to 196±43 bpm in AF, and the RV EDV decreased from 70±18 ml in SR to 63±16 ml in AF, while there was no significant difference in RA pressure. Thus, it seems that the experimental acute AF in the present study resulted in a rapid ventricular response, reflecting only a limited aspect of the various hemodynamic consequences of AF on atrial/ventricular/annular dynamics. Although I acknowledge that the authors provided some explanation regarding this issue in the Limitations section, I think it needs further elaboration in the Discussion section.

Author's Responses to Questions:

We fully agree with the Reviewer. The methodology used (experimental AF) may not fully reproduce all hemodynamic changes observed in all clinical scenarios. Clinically, in acute atrial fibrillation, the heart rate can vary widely, but it is typically elevated. We believe that our results describe this most common situation. 

We have extended the Limitation section of the manuscript to fully acknowledge that. 

#2 Reviewer’s comments [3]: The lack of measurement of RA size (diameter or volume), the lack of significant difference in RA pressure between SR and AF, and the reduced RV EDV under rapid atrial pacing (experimental AF) need to be further emphasized in the Discussion or Limitations section. Considering that the dilation of the RA and RV, as well as the increase in RA pressure, are known to be the main drivers of the development and progression of tricuspid regurgitation, these points are crucial. It could also be assumed that the lack of detectable TR in the authors’ ovine experimental model could be due to the experimental design, which induced a rapid ventricular response but did not cause dilation of the RA and RV.

Author's Responses to Questions:

Indeed, the lack of measurements regarding the right atrium raises some concerns and questions. However, our work was intended to focus on changes in geometry and strains in the tricuspid valve annulus in the acute settings. We believe that the experimental design , so it was not extended to the right atrium. The lack of data is not an escape from questions, but only an assumption made before the study began.

[2]: Discussion section (page 15, lines 292 – 294): The sentence, “Physiologically, coordinated atrial contraction reduces the annulus before ED to prepare the valve for closure with subsequent ventricular systole completing valve closure”, seems a bit confusing. Given the contents of the referenced article in this sentence (Tsakiris AG et al. Circ Res. 1975;36:43-8), I wonder whether the authors aimed to describe the reduced annular size (or area) during coordinated atrial contraction.

Author's Responses to Questions: 

Based on the cited reference (Tsakiris AG, et al. . Motion of the tricuspid valve annulus in anesthetized intact dogs. Circ Res. 1975 Jan;36(1):43-8. doi: 10.1161) we in fact wanted to describe the pre-systolic annular reduction during atrial contraction. We slightly reworded the sentence that reads now: “ Physiologically, coordinated atrial contraction reduces the annular size before ED to prepare the valve for closure26 ,with subsequent ventricular systole completing valve closure. The degree of this pre-systolic annular contraction was found to be lower in AF than in SR in our study, and this physiologic mechanism, when disturbed in AF, could be associated with initiation and progression tricuspid insufficiency in patients with long-standing AF.” 

[4]: Figure 6: I recommend the authors add indications for the orientation of the tricuspid annulus (i.e., Anterior, Septal, and Posterior) in the figure, as done in their previous publication (Figure 4 in the study by Malinowski et al. J Thorac Cardiovasc Surg 2019;157:1452-61). The current indications (#1, #3, and #5) are sufficient to find the orientation but could be confusing for some readers if additional descriptions (i.e., Anterior, Septal, and Posterior) are not provided.

Author's Responses to Questions:

Thank you for your suggestion. Corrections have been made to Figure 4.  

[5]: Were there any specific reasons for describing the tricuspid annular area change (contraction) without providing a decimal place? It seems a bit awkward as most other measurements are provided to the first decimal place, but the “tricuspid annular area change” values are not.

Author's Responses to Questions:

Indeed, reporting area results without a decimal place is inconsistent. Corrections have been made in the manuscript. Thank you for pointing this out. Now the tricuspid annular area change is reported with a decimal place. 

[6]: Minor comments

• Abstract: Please review the abbreviation for AF in the Abstract (Results section: Atrial fibrillation perturbed systolic global annular strain…).

• Abstract: Please check the number of standard derivation for the post-procedural systolic global annular strain (-2.78±-2.88%).

• Methods: In the Methods section of the main text, please use a consistent form of describing the states of the USA for devices/software used in the present study (i.e., Medtronic, Minneapolis, “MN”; Konigsberg Instruments Inc, Pasadena, “Calif”; MathWorks, Natick, “Mass”; GraphPad Software, San Diego, “CA”).

• In the main text, “Figure 3 (page 7)”• Discussion (page 14): When citing the study by DH Kim et al. (Ref #19: JACC Cardiovasc Imaging. 201 appears earlier than “Figure 2 (page 9)”.

• I recommend the authors use a consistent form for describing the p-value (in some figures, the p-values are described as 0.XXX, but in other figures as .XXX).

• Please review the abbreviations for SR and AF throughout the manuscript (they appear in their abbreviated forms in the Results section [page 12 – Annular Strains], but in full names in other parts of the manuscript). Additionally, please check the abbreviation for the tricuspid annulus and functional tricuspid regurgitation, as they sometimes appear in their abbreviated forms (TA and FTR), but sometimes in their full name (tricuspid annulus, functional tricuspid regurgitation, or functional TR).

9;12:665-677), please use the family name of the author (“Kim”), rather than the given name (“Dae-Hee”).

Author's Responses to Questions:

Thank you for your above suggestions and comments. We have corrected all the mistakes pointed out and corrected the manuscript as suggested.

---

## [Editor Report · Decision Letter 1]

17 Sep 2024

Ovine tricuspid annular dynamics and three-dimensional geometry during acute atrial fibrillation

PONE-D-24-14958R1

Dear Dr. Marcin Malinowski

We’re pleased to inform you that your manuscript has been judged scientifically suitable for publication and will be formally accepted for publication once it meets all outstanding technical requirements.

Kind regards,

Ibrahim Marai, MD

Academic Editor

PLOS ONE

---

## [Editor Report · Acceptance letter]

24 Sep 2024

PONE-D-24-14958R1 

PLOS ONE

Dear Dr. Malinowski, 

I'm pleased to inform you that your manuscript has been deemed suitable for publication in PLOS ONE. Congratulations! Your manuscript is now being handed over to our production team.

Kind regards, 

on behalf of

Dr. Ibrahim Marai 

Academic Editor

PLOS ONE